Uncertainties regarding the natural mortality of fish can increase due global climate change

Pereira Campos Caroline 1
Bitar Sandro Dimy Barbosa 2
http://orcid.org/0000-0001-5406-0998 Freitas Carlos 3 cefreitas@ufam.edu.br
1 Instituto Federal de Roraima , Caracarai, Roraima , Brazil
2 Departamento de Matemática, Universidade Federal do Amazonas , Manaus, Amazonas , Brazil
3 Departamento de Ciências Pesqueiras, Universidade Federal do Amazonas , Manaus, Amazonas , Brazil
Zucchetta Matteo
Electronic publication date: 2023 Mar 7
Publication date: 2023
Volume: 11
Electronic Location ID: e14989
Received 2022 Aug 24; Accepted 2023 Feb 12
Copyright: © 2023 Pereira Campos et al.
Copyright year: 2023
Copyright holder: Pereira Campos et al.
License: This is an open access article distributed under the terms of the Creative Commons Attribution License, which permits unrestricted use, distribution, reproduction and adaptation in any medium and for any purpose provided that it is properly attributed. For attribution, the original author(s), title, publication source (PeerJ) and either DOI or URL of the article must be cited.
License URL: https://creativecommons.org/licenses/by/4.0/

Keywords: Climate change, Natural mortality of fish, Fuzzy theory, Uncertainty, Life strategies of fish

Funding: Coordenação de Apoio de Pessoal de Nível Superior—CAPES This work was supported by Coordenação de Apoio de Pessoal de Nível Superior—CAPES. The funders had no role in study design, data collection and analysis, decision to publish, or preparation of the manuscript.

==============================
The increase in temperature resulting from global climate change can directly affect the survival of fish and therefore population parameters such as natural mortality (M). The estimation of this parameter and the understanding of the uncertainties in its estimates are enormous challenges for studies that evaluate fish stocks. In addition, the effects of increases in temperature may be associated with life strategies. Therefore, the fuzzy set theory was used to evaluate the effects of temperature increase on the natural mortality of fish, considering different life strategies. The model showed that the increase in temperature increased the uncertainties in M estimates for all species, regardless of the life strategy. However, opportunistic species present greater uncertainties in estimates of M compared to equilibrium species. The patterns found in uncertainties of M associated with species groupings by life strategies can be used in holistic approaches for the assessment and management of recently exploited fisheries resources or for those with limited biological data.

Introduction

Climate change is considered as one of the biggest and most complex environmental and socioeconomic problem of the world at the present time (Pörtner et al., 2022). Global warming is the result of natural alterations in the climate system and human activities, which contribute to rising levels of greenhouse gas (GHG) emissions (Pörtner et al., 2022). Scientific evidence indicates that anthropogenic emissions of GHGs are the main cause of the increase in temperature and the biggest culprit in relation to the current environmental imbalance, causing more than half of the observed increase in global average surface temperatures of the planet (Cleugh et al., 2011). The temperature increases on the surface of the planet are among the phenomena with greater predictability and with wide-ranging effects (Pörtner et al., 2022).

Temperature is an important environmental variable, and it is directly related to the survival of many ecothermic animals, especially fish (Baudron et al., 2014). The effects on fish populations can be manifested through changes in abundance, migration patterns and geographical distribution (Baudron et al., 2014; Ficke, Myrick & Hansen, 2007). Oremus et al. (2020) made projections using global climate change scenarios in order to verify changes in the migration and distribution of 779 species of fish from each exclusive economic zone (EEZ) and concluded that tropical countries are more vulnerable to a potential loss of stocks of these species, since these migrate to cooler waters to maintain their thermal comfort.

There are patterns that correspond to different life strategies of freshwater and marine fish species (Winemiller, 1989; Winemiller & Rose, 1992; Röpke et al., 2017). The plasticity of species with different life strategies that depend on environmental variations has also been widely studied (Winemiller & Rose, 1992). Researchers have been indicating that species with life strategies characterized as having a large size, high longevity, low growth and low mortality rate are at a disadvantage in terms of temperature increase, due to their higher metabolic rate, and will likely face local extinction (Freitas et al., 2013). On the other hand, smaller species in lower trophic levels can benefit from this potential reduction of predators (Woodward et al., 2016). This shows that the effects of climate change on fish can be of a greater or lesser degree and can have distinct signs, depending on the life strategy of each species (Blanck & Lamouroux, 2007; Röpke et al., 2017).

The natural mortality M is one of the most important parameters in the assessment of fish stocks and it is essential in order to estimate the productivity of a fish stock, but it is also one of the most difficult to estimate, due to limitations of data (Punt et al., 2021). The magnitude of error, bias and variance in the estimation of this parameter can be substantial and can be affected by temporal and spatial effects (Kenchington, 2014). Incorporating the uncertainties in the calculation of M estimates is essential to make robust assessments of the fish stock (Brodziak et al., 2011).

Most estimation methods for M consider only intrinsic factors of the species: length, maturation size, growth rate and age, without considering the influences of extrinsic factors, such as environmental variables (e.g., temperature) (Kenchington, 2014), which can be determinants in population dynamics and are intrinsically variable (Sæther & Bakke, 2000). The method proposed by Pauly (1980) to estimate M includes intrinsic and extrinsic factors, since it estimates M as a function of the von Bertalanffy (1938) growth parameters and the average annual water temperature. It has been shown that the higher the temperatures, the higher the natural mortality rates in fish (Sparre & Venema, 2007).

However, the effects of temperature increase, due to climate change, incorporate high variability, which is in part due to the uncertainties existing in the climate models and underlying scenarios (Jeppesen et al., 2014). Thus, it is necessary to apply methodological approaches that incorporate variations and uncertainties in the estimation of parameters. In this sense, fuzzy logic and fuzzy set theory are presented as appropriate tools in the modeling of biological phenomena (Barros, Bassanezi & Lodwick, 2017). This theory was introduced by Zadeh (1965), with the idea of contrasting deterministic models to more flexible models, which enable mathematical modeling of uncertainty, besides mathematically formulating the subjectivity inherent to natural phenomena. Fuzzy logic has been used as a tool in several issues related to the fisheries, such as assessing the vulnerability of marine species (Cheung, Pitcher & Pauly, 2005), assessment of pelagic fishery ecosystems (Paterson et al., 2007), classification of fishing areas (Sylaios, Koutroumanidis & Tsikliras, 2010), spatial management of marine fisheries (Teh & Teh, 2011), behavior of fishers in pelagic fisheries (Wise et al., 2012), stock-recruitment relations (Gutiérrez-Yurrita, 2014), parameter estimates of the relationship between weight and length (Bitar, Campos & Freitas, 2016), dynamic of competition among species taking into account abiotic factors and fishing effort (Souza & Bassanezi, 2018), and the fuzzy interactivity in the generalized model of von Bertalanffy (Souza & Prata, 2019).

In this study, we assumed two points: (i) the estimate of the average annual water temperature, calculated as a fuzzy number, could, therefore, allow an estimate of natural mortality that incorporates uncertainties in the face of global climate changes; and (ii) the amplitude and intensity of the effects of environmental phenomena could be species-specific (Röpke et al., 2017). We then answered three questions: (i) Will the temperature increases, in the scenarios proposed by the IPCC, increase uncertainties regarding the natural mortality in fish? (ii) Are these uncertainties, if noticeable, the same for marine and freshwater fish species? (iii) Are these uncertainties, if noticeable, the same for fish species with different life strategies?

Materials and Methods

Study area

The area covered in this study was the Brazilian tropical region, including continental and maritime areas (Fig. 1). The main feature of the climate of the tropics is the high average monthly temperatures. However, factors such as latitude, altitude, circulation of air masses and global climate change can cause climate differentiation.

Figure 1 Map of the Brazilian tropical zone.

The zones indicates watersheds and temperature collection points (river monitoring stations).

Classification of life strategies of the target species

As an alternative to the concepts of r and k strategists (Pianka, 1970) and based on mechanisms of trade-offs between reproduction, growth and survival of freshwater fish, Winemiller (1989) proposed a classification using three categories of life strategies: equilibrium, opportunistic and seasonal. The classification proposed by Winemiller (1989) gave rise to the triangular continuum of strategies model, which resulted from adaptive responses to environmental predictability and variability. Subsequently, Winemiller & Rose (1992), while studying freshwater and marine fish, confirmed this classification. Basically, these patterns of life strategies suggest that:

(1) Equilibrium strategists—These are fish with low growth rates and natural mortality, with a large body size, late first maturation, split spawning, large oocytes, low fecundity, broad reproductive period, relatively high longevity, parental care, which produce larger offspring in low quantities. Since they produce relatively few offspring, survival from the early stages must be high for these populations to persist. These are species that live in relatively stable environments.

(2) Seasonal strategists—These are fish that exhibit migratory behavior, absence of parental care, growth rates and natural mortality from intermediate to high, with intermediate to large body size, first maturation is intermediate to late, small oocytes, high fecundity and short periods of reproduction. This strategy would be associated with both temporal and seasonal variations and, if these conditions are favorable, a synchronization in reproduction will occur, guaranteeing juvenile individuals greater chances of growth and survival, in addition to providing a rapid population replacement.

(3) Opportunistic strategists—These are fish with a small body size, early first maturation, small oocytes, low fecundity, prolonged breeding period, short life span and with little or no parental care. These characteristics are related to the ability of rapid colonization of environments marked by unpredictable temporal variations.

Species selected for the study

Species representing the different life strategies were selected according to the classification of Winemiller (1989) and Winemiller & Rose (1992). In addition, we sought to select species from freshwater and marine environments that have been fished for commercial purposes. Based on these criteria, three freshwater and three marine species were selected, each representing a particular type of life strategy. The growth parameters (L∞ and k), of these species can be observed in Table 1.

Table 1 Population parameters of the species selected for the study, where: L∞ = theoretical maximum length; k = individual growth coefficient; L50 = first maturation; Tmáx = longevity.

Species	L∞ (cm)	k (year−1)	L50 (cm)	Tmáx (year)	Strategy	Environment	
Cichla temensis	68.05	0.20	31.11	14.00	Equilibrum	Freshwater	
Prochilodus nigricans	34.60	0.44	23.08	6.80	Seasonal	
Triportheus angulatus	26.78	0.77	16.30	3.89	Opportunist	
Cynoscion acoupa	142.90	0.13	42.07	10.00	Equilibrum	Marine	
Lutjanus synagris	56.00	0.22	23.00	18.00	Seasonal	
Sardinella brasiliensis	23.68	0.26	15.80	10.73	Opportunist	
Note:

Sources : 1—Campos, Freitas & Amadio (2015), Campos, Catarino & Freitas (2020); 2—Catarino et al. (2014), Camargo & Lima (2008); 3—Prestes et al. (2010); 4—Oliveira et al. (2020); 5—Aschenbrenner et al. (2017); 6—Costa, Tubino & Monteiro-Neto (2018).

Species of freshwater environments:

Equilibrium Strategist: Cichla temensis (Humboldt, 1821)

Order: Perciformes. Family: Cichlidae

The peacock bass, Cichla temensis, has its distribution restricted to the blackwater rivers and their tributaries in the basins of the Amazon River and Orinoco River. It has great importance in the diets of the riparian population, as well as in the ornamental fish trade and, especially, in sport fishing (Soares et al., 2007). In 2003, the peacock bass represented 2.58% of the fishing production that was landed in the main ports of the state of Amazonas (Ruffino et al., 2006) and in the middle Negro River region, and it was the third most landed fish in 2013 (Inomata & Freitas, 2015). It is characterized by a preference for lentic environments, edges of lakes and sandbanks in the main channel of rivers (Winemiller, Taphorn & Barbarino-Duque, 1997). Cichla temensis is a predator at the top of the chain, with a piscivorous habit and it differs among the 15 species of the genus Cichla, since it reaches larger sizes, which can be up to 80 cm and more than 11 Kg (Winemiller, Taphorn & Barbarino-Duque, 1997; Kullander & Ferreira, 2006). The couple builds the nest and, one or both, take care of the offspring (Zaret, 1980; Jepsen, Winemiller & Taphorn, 1999). It presents split spawning, with a long reproductive period and low fecundity (Jepsen, Winemiller & Taphorn, 1999). We used these natural history data to approximate model parameters for major life history characteristics for each of the target species. We considered C. temensis to have low growth rates (k) and natural mortality (M), high values of theoretical maximum lengths (L∞) and first maturation (L50), as well as relatively high longevity (Tmax).

Seasonal Strategist: Prochilodus nigricans (Spix & Agassiz, 1829)

Order: Characiformes. Family: Prochilodontidae

The curimatã, Prochilodus nigricans, is found in the Amazon River and Tocantins River basins (Castro & Vari, 2003). It has significant commercial value and is one of the most caught species in Brazilian continental waters. It is predominant in the landings in Amazonian cities (Batista et al., 2012) in addition to its importance in subsistence fishing (Isaac et al., 2015; Zacarkim et al., 2015). This species inhabits large rivers, flooded forests, lakes and streams, and can be found at any time of the year, even if its abundance varies, due to the influence of seasonal fluctuations in the level of the rivers in the Amazon (Silva & Stewart, 2017). It performs trophic, reproductive and dispersal migrations throughout the year and feeds, basically, on organic debris and periphyton (Silva & Stewart, 2017). Like other migratory fish that feed on detritus, it has access to an abundant source of energy, which contributes to the modulation of carbon flow and productivity of ecosystems (Silva & Stewart, 2017). It is a medium-sized species, between 35 and 50 cm long and is around 3 Kg in weight (Santos, Ferreira & Zuanon, 2006). It has high fecundity, small oocytes and short reproductive period (at the beginning of the rising water period) and does not provide parental care (Silva & Stewart, 2017). In addition, it presents high intermediate values of population parameters, such as growth constant (k), theoretical maximum length (l∞), natural mortality (M) and first maturation length (L50). Röpke et al. (2017) also classified P. nigricans as a seasonal strategist.

Opportunistic Strategist: Triportheus angulatus (Spix & Agassiz, 1829)

Order: Characiformes. Family: Characidae

A total of 16 species have been described for the genus Triportheus, in which three are well known, Triportheus albus (Cope, 1872), T. auritus (Valenciennes, 1850) and T. angulatus (Spix & Agassiz, 1829), and their occurrences are recorded in the basins of the Amazon, Tocantins and Orinoco Rivers (Malabarba, 2004). In recent years, the demand and commercial interest for Amazonian fish species that are numerous, but of smaller size, has increased significantly. For example, in the 1970s, sardines were infrequently commercialized (Petrere Jr., 1978), but in the past decade, there has been an increase in the volume of landings in the ports of Manaus and Manacapuru, Amazonas state, which has risen from 2% to 12% (Gonçalves & Batista, 2008). Triportheus angulatus is a pelagic fish that commonly lives near the surface and close to the banks of rivers and lakes. It has an omnivorous diet that includes insects, zooplankton, fruits and seeds, and it has been verified that, even under conditions of low oxygen concentrations, its feeding activity is not influenced as it has the ability to modify its lower lips to absorb surface oxygen from water (Yamamoto, Soares & Freitas, 2004).

Although T. angulatus does not meet some characteristics to be assumed as a typical opportunistic species, it is one of the exploited species by fishing fleets in freshwater systems that shows more characteristics of this type of life strategy, such as small body size, ranging between 20 and 24 cm (Malabarba, 2004); low fecundity and a single spawning (Araújo et al., 2012); no parental care and high growth (k) and natural mortality (M) rates, low value of theoretical maximum lengths (L∞), small first maturation lengths (L50) and low life span (Tmax).

Species of marine environments:

Equilibrium Strategist: Cynoscion acoupa (Lacepède, 1801)

Order: Perciformes. Family: Sciaenidae

The red mullet, Cynoscion acoupa, has a wide distribution and occupies coastal waters from Panama, in Central America, to Argentina, in southern Latin America as well as most of the Brazilian coast (except in the extreme south of the country), both in marine and estuarine environments (Cervigón, 1993). Cynoscion acoupa is the third most landed fish species in Brazil and has a high commercial value due to the quality of its meat and for its swim bladder, which is used as a raw material in the production of emulsifiers and clarifiers (Barletta, Barletta-Bergan & Saint-Paul, 1998). Cynoscion acoupa is a species with nectonic, demersal and coastal habits that lives in shallow and brackish waters under the influence of estuaries and mangroves, which, for this species, are places of refuge, feeding and reproduction (Szpilman, 2011). It feeds mainly on fish and crustaceans and is a large species that can reach up to 130 cm in length and 20 Kg in weight (Ferreira et al., 2016). It has batch spawning, a long reproductive period and high fecundity (a common characteristic for marine species in equilibrium) (Almeida et al., 2016). It has low growth rates (k) and high values of theoretical maximum lengths (L∞) and first maturation lengths (L50). Its longevity is considered as intermediate to high (Tmáx).

Seasonal Strategist: Lutjanus synagris (Linnaeus, 1758)

Order: Perciformes. Family: Lutjanidae

The lane snapper, Lutjanus synagris, has an area of occurrence that extends from North Carolina (USA) to southeastern Brazil (Manooch & Mason, 1984). It is an important fishing resource of tropical marine waters, due to its abundance and quality of meat, which gives it high commercial value and, as a result, it is the target of commercial, artisanal and sports fisheries (Rezende, Ferreira & Frédou, 2003). Lutjanus synagris is a species of demersal habits that lives in warm waters, and is often associated with rocky and coral bottoms, between the coastal zone and out to depths of about 400 m (Pimentel & Joyeux, 2010). It feeds mainly on crustaceans and fish. Considered a medium-sized species, it can reach lengths of between 40 and 50 cm (Pimentel & Joyeux, 2010). It has batch spawning and high fecundity (Grimes, 1987; Freitas et al., 2014). It presents intermediate to slow growth (k), as well as intermediate values of theoretical maximum lengths (L∞) and first maturation (L50), and is relatively long-lived (Tmax). King & McFarlane (2003) concluded that species of the family Lutjanidae are seasonal strategists.

Opportunistic Strategist: Sardinella brasiliensis (Steindachner, 1879)

Order: Clupeiformes. Family: Clupeidae

The Brazilian sardinella, Sardinella brasiliensis, is a species that is endemic to the southeastern coast of Brazil, and is found along the continental shelf, between Cabo de São Tomé, RJ and Cabo de Santa Marta Grande, SC (Paiva & Motta, 2000). In terms of volume of production, it is the most important marine fishing resource in Brazil since it is the most commercialized and consumed fish in the country (Paiva, 1997). Shoals occur near the surface, in water up to 80 m deep, and reduce their frequency as the depth increases (Paiva & Motta, 2000). In the region where it occurs, specific oceanographic characteristics are observed, due to the periodic entry of infiltrations from the Central South Atlantic water. These waters act as a fertilization mechanism and provide a greater amount of plankton that favors the survival of the larvae of S. brasiliensis (Matsuura, 1998). It is a species of planktophagous feeding habits (Kurtz & Matsuura, 2001), of small size, ranging from 9 to 27 cm, with a high fertility rate, sequenced spawning and without parental care (Costa, Tubino & Monteiro-Neto, 2018). In addition, it also presents a high natural mortality rate (M) and early maturation (L50), with low values for theoretical maximum lengths (L∞). Costa, Tubino & Monteiro-Neto (2018) also classified S. brasiliensis as an opportunistic strategist.

Data collection

To estimate the fuzzy set of natural mortality (M) we use: (i) input variables for the natural mortality equation (M) proposed by Pauly (1980): von Bertalanffy growth parameters (Table 1) of species and annual average water surface temperature (National Water Agency of Brazil (ANA), 2020); and (ii) environmental variable in the modeling: temperature estimates proposed by the Intergovernmental Panel on Climate Change (IPCC, 2014).

Average annual temperature

The National Water Resources Information System (SNIRH), whose organization, implementation and management are the responsibility of the National Water Agency, divides the national territory into eight large basins: 1—Amazon River, 2—Tocantins River, 3—Atlantic (northern/northeastern stretch), 4—São Francisco River, 5—Atlantic (eastern stretch), 6—Paraná, 7—Uruguay, and 8—Atlantic (southeastern stretch) (National Water Agency of Brazil (ANA), 2020). The temperature data concerning the basins 1, 3 and 5 (Fig. 1), where the studies of estimation of growth parameters of the species were carried out (Table 2), were obtained and used in the modeling.

Table 2 Area of study of the target species and their respective basins of origin.

Study No	Study area	Basin	
1	Negro River	1—Amazon River	
2	Solimões River	1—Amazon River	
3	Solimões River	1—Amazon River	
4	Baía de São Marcos	3—N/NE Atlantic	
5	Banco dos Abrolhos	5—Eastern Atlantic	
6	Costa do Itaipu	5—Eastern Atlantic	
Note:

Sources : 1—Campos, Freitas & Amadio (2015), Campos, Catarino & Freitas (2020); 2—Catarino et al. (2014), Camargo & Lima (2008); 3—Prestes et al. (2010); 4—Oliveira et al. (2020); 5—Aschenbrenner et al. (2017); 6—Costa, Tubino & Monteiro-Neto (2018).

IPCC scenarios

In 2014, the Intergovernmental Panel on Climate Change (IPCC, 2014) proposed four more simplified scenarios, but using a more complete system, known as RCPs (representative concentration pathways), which use the amount of energy absorbed (radiative forcing, in W/m2) by greenhouse gases (GHG), during or near the end of the 21st century. RCP 2.6: mitigation scenario, which leads to a very low level of absorption; RCP 4.5 and RCP 6.0: stabilization scenarios and RCP 8.5: scenario with very high greenhouse gas emissions (Table 3) (IPCC, 2014).

Table 3 Features of the RCP scenario of the fifth IPCC report (AR5).

RCP	Radioactive forcing	Concentration CO2-equiv. (ppm)	Behavior	Rise in sea level (cm)	Elevation of T°C on the planet in 2,100	
2.6	Peak of <3 W.m−2 in 2,100	490	Rising with a peak in 2,040 and declining	26–55	0.3–1.7	
4.5	Additional storage of ≈4.5 W.m−2	650	Rising until 2,060 and stabilizing	32–63	1.1–2.6	
6.0	Additional storage of ≈6 W.m−2	850	Rising until 2,100 and stabilizing	33–63	1.4–3.1	
8.5	Storage of ≈8.5 W.m−2	>1370	Rising until 2,100	45–82	2.6–4.8	
Note:

Source: IPCC (2014).

Data analysis

Foundations of the fuzzy set theory

To understand the modeling process presented in this study, it is necessary to define concepts related to the theory of fuzzy sets. All concepts were defined according to Barros, Bassanezi & Lodwick (2017) and are detailed in Appendix S1.

Estimation of fuzzy natural mortality

The fuzzy set for natural mortality of each species was performed in four stages:

(i) Fuzzy set estimate for average annual temperature:

For the fuzzification of average annual temperature, the normality of the data distribution was first verified using the Shapiro–Wilk normality test and its variance. Then, the average annual temperature was modeled as a fuzzy number, from a set of confidence intervals, based on the methodology of Buckley (2005).

Buckley (2005) suggests finding the confidence interval 100 (1 − β)% for all 0.01 ≤ β < 1. Each of these intervals can be denoted by: [θ1(β), θ2(β)].

The range is considered [θ*, θ*] for β = 1, where θ* it is the point estimate for the parameter θ, thus, there are intervals for 0.01 ≤ β ≤ 1. These ranges define the fuzzy set θ^ through its α-levels as follows: [θ^]α=[θ1(α),θ2(α)].

For 0 ≤ α < 0.01, [θ^]α=[θ1(0.01),θ2(0.01)] was defined. With this, from the confidence intervals for a parameter θ, a fuzzy set for the average annual temperature was constructed whose α-cuts are those intervals. The fuzzy average annual temperature was estimated for each set of temperature data, according to the basin of origin in the study area for each species.

(ii) Fuzzy modeling for the temperature variations suggested by the IPCC ( Δ~i)

For each IPCC scenario, a fuzzy average annual temperature was projected. The membership functions, used for each IPCC scenario, were assumed to be the characteristic function of the set of each temperature range, since it is believed that all values of this range have the same importance, since there is no evidence of differences.

For each interval Δi=[Δmi,ΔMi], with (i=1,2,3,4) and where Δmi is the minimum value and ΔMi is the maximum value in the IPPC interval, associated with the temperature variation suggested by the IPCC, a fuzzy set Δ~i is assigned. Therefore, a membership function is defined to characterize this fuzzy set. We will use the characteristic function (or indicator function) of each set Δi. With this configuration, the α-levels of such sets are [A]α=Δi, 0≤α≤1. Algebraic operations on these fuzzy sets are performed on their α-levels, based on interval arithmetic (Moore, 2009). These fuzzy sets will be used to estimate the average temperature variation under the influence of variations predicted by the IPCC.

(iii) Membership function for average annual temperature values under IPCC scenarios ( T~i, i=1,2,3,4)

Let Δ~i the diffuse variation, associated with scenario i, obtained with the methodology presented above. The fuzzy set estimate for average temperature Ti~, relative to scenario i, will be calculated with the sum of two fuzzy numbers, acocording to the following equation:

Ti~=T~⊕Δ~i, where ⊕ denotes sum of fuzzy numbers (Barros, Bassanezi & Lodwick, 2017).

It is noteworthy that, (Theorem 2.4, Barros, Bassanezi & Lodwick, 2017)

[Ti]~α=[T~]α+[Δ~i]α

and, by Theorem 1.4 in Barros, Bassanezi & Lodwick (2017), it is possible to characterize T~i through its α-levels.

(iv) Fuzzy set for natural mortality

In each scenario, the fuzzy natural mortality was estimated by applying the Zadeh’s Extensions Principle (Zadeh, 1965) of the classical function proposed by Pauly (1980):

(1) ln⁡(M)=−0.0152−0.279ln⁡(L∞)+0.6543ln⁡(k)+0.463ln⁡(T)

where L∞, k and T denote theoretical maximum length, individual growth coefficient and average annual water surface temperature, respectively.

The procedures for calculating the Zadeh extension are presented in the pseudo code, detailed in Appendix S2. We coded our simulations in Python programming language, version 3.10.6, along with matplotlib, scipy, numpy and pandas packages, on a computer equipped with an Intel(R) Core (TM) i5-2450M and 2.50 GHz CPU. The four stages are contained in pseudocode, and documented in Appendix S2.

Results

The fuzzy sets for the average annual temperature for the Amazon River basin have a temperature range between 24 °C and 32 °C (Fig. 2A), while the northern/northeastern Atlantic basin has a temperature range of between 26 °C and 33 °C (Fig. 2B) and the eastern Atlantic basin ranges between 19 °C and 30 °C (Fig. 2C). Taking into account the highest degree of membership, that is, 1 (one), the average temperature is 27.3 °C, 28.8 °C and 25.3 °C, for the Amazon River, northern/northeastern and eastern basins, respectively.

Figure 2 Fuzzy annual average temperature.

(A) Amazon River, (B) Northern—Eastern Atlantic and (C) Eastern Atlantic. Each curve was obtained as fuzzy set to represent the interval of values of temperature.

For the three freshwater species, a pattern of increased uncertainty of M with increasing temperature was observed. This uncertainty is most evident between scenarios RCP 2.6 and RCP 8.5. Cichla temensis, which presents strategy in equilibrium, presented the lowest degree of uncertainty of M among the freshwater species (Fig. 3A). Among these species, Triportheus angulatus, that presents some characteristics of opportunistic species, presented the highest degree of uncertainty of M for all scenarios (Fig. 3B). Prochilodus nigricans, a seasonal species, presented an intermediate degree of uncertainty when compared to other freshwater species (Fig. 3C).

Figure 3 Distribution of possibilities of natural mortality (M) of freshwater species.

(A) Cichla temensis, (B) Prochilodus nigricans and (C) Triportheus angulatus. The vertical bar on the right indicates the degree of membership for each IPCC (Intergovernmental Panel on Climate Change) scenario, after the application of the Zadeh extension. The vertical axis on the left presents the possibilities of M. The bar on the right, in grayscale, presents the degree of membership, and the closer to 1 (one), the greater the pertinence of M.C1 = RCP 2.6; C2 = RCP 4.5; C3 = RCP 6.0; C4 = RCP 8.5 (IPCC, 2014).

In the same manner as was presented for the freshwater species, the observed pattern is an increase in the uncertainty of M with the increase in temperature for the three saltwater species, and this is more evident between scenarios RCP 2.6 and RCP 8.5. When the life strategies were compared, Cynoscion acoupa, an equilibrium strategist, presented the lowest degree of uncertainty between the species (Fig. 4A). This increased for Lutjanus synagris (Fig. 4B), which is a seasonal strategist, and Sardinella brasiliensis, an opportunist, presented the highest degree of uncertainty of M, among the species (Fig. 4C). Considering a degree of relevance greater than or equal to 0.05, the α-cut associated with the distribution of the possibility of M for each species and for each scenario of the IPCC can be seen in Table 4.

Figure 4 Distribution of possibilities of natural mortality (M) of marine species.

(A) Cynoscion acoupa, (B) Lutjanus synagris and (C) Sardinella brasiliensis. The vertical bar on the right indicates the degree of membership for each IPCC (Intergovernmental Panel on Climate Change) scenario, after the application of the Zadeh extension. The vertical axis on the left presents the possibilities of M. The bar on the right, in grayscale, presents the degree of membership, and the closer to 1 (one), the greater the pertinence of M.C1 = RCP 2.6; C2 = RCP 4.5; C3 = RCP 6.0; C4 = RCP 8.5 (IPCC, 2014).

Table 4 Natural mortality possibilities (M) for each target species in the study and IPCC (Intergovernmental Panel on Climate Change) scenarios, considering a degree of relevance greater than or equal to 0.05.

Species	Current	RCP 2.6	RCP 4.5	RCP 6.0	RCP 8.5	Strategy	Environment	
Cichla temensis	0.48	0.49–0.50	0.51	0.49–0.51	0.50–0.53	Equilibrium	Freshwater	
Prochilodus nigricans	0.98–0.99	0.99–1.02	1.00–1.03	1.01–1.04	1.03–1.07	Seasonal	
Triportheus angulatus	1.52–1.54	1.53–1.58	1.55–1.60	1.56–1.62	1.59–1.66	Opportunist	
Cynoscion acoupa	0.30	0.31	0.31	0.31	0.33	Equilibrium	Marine	
Lutjanus synagris	0.52–0.53	0.52–0.55	0.53–0.56	0.54–0.56	0.55–0.58	Seasonal	
Sardinella brasiliensis	0.74–0.76	0.75–0.78	0.76–0.79	0.76–0.80	0.78–0.82	Opportunist	

It was possible to identify a general pattern associated with estimates of M, which resulted from IPCC scenarios, and two patterns related to the interactions between the life strategy and the type of environment inhabited by the species. The uncertainty of M increases gradually from one scenario to another, and is consistently higher in RCP 8.5 for all species. However, the variation in the uncertainty is not constant between species with different life strategies. Species in equilibrium have lower uncertainty in estimates of M in all scenarios. Seasonal species present estimates of M with intermediate values of uncertainty, while opportunistic species or with characteristics that come close to the opportunistic strategy, such as T. angulatus, exhibit the greatest uncertainties in the estimated M values for all scenarios, and the variation in the increase in natural mortality is slightly greater for these species. Apparently, species that inhabit freshwater systems have a higher degree of uncertainty of M when compared to marine species.

Discussion

The difficulty in estimating natural mortality is associated with the number of factors that influence it and the complexity of taking into account all or most of these factors in the estimation process. To minimize this problem, Brodziak et al. (2011) suggested considering the relative influence of intrinsic vs. extrinsic factors and using the estimated distribution of M to provide a more accurate approximation of parametric uncertainties. Our model used temperature as a key factor, which can influence the ratio of metabolic rate and body mass as an intrinsic factor, as well as an environmental and extrinsic factor. Therefore, this can be considered a more robust model compared to the models that have already been used in the attempt to incorporate the uncertainties about M, but that do not combine the two groups of factors that affect M or do not use a force variable that can influence the intrinsic and extrinsic factors (MacCall, 2011; Punt et al., 2021).

In a recent study, Dahlke et al. (2020) analyzed 694 freshwater and marine fish species from all climate zones and concluded that 60% of fish species may not survive if climate warming reaches the worst-case scenario, i.e., increasing to around 4 °C. However, for a more optimistic scenario, with an increase of up to 1.5 °C, only 10% of the species surveyed would be at risk in the next 80 years. The physiology of fish is directly related to temperature, which has an inverse relationship with the amount of dissolved oxygen in the aquatic environment (Ficke, Myrick & Hansen, 2007). This inverse relationship (increase in temperature and the reduction of oxygen level) led to an increase in metabolic rate, an increase in physiological stress, and an imbalance in biochemical functions, as well as an increase in the mortality rate of eggs, larvae and adult fish, thus changing the structure, composition and population dynamics of the fish species (Freitas et al., 2013; Barros & Albernaz, 2013). Moreover, the natural mortality is related to body length, asymptotic length, and growth rate (Gislason et al., 2010).

The changes in fish populations that are induced by climate change seem to be more noticeable in smaller species with a higher growth rate (Levangie, Blanchfield & Hutchings, 2021). These authors investigated the effects on M of a 10% reduction in the asymptotic length of marine fish species of Canada’s Scotian shelf, associated with a 1 °C increase in ocean temperature, and found that species with smaller body size suffered greater increases in M when compared to species with a larger body size. Additionally, considering the 1 °C increase in ocean temperature, Wang et al. (2020) showed that small-sized, fast-growing species were more likely to experience declines in population growth than larger, slow-growing species. These studies corroborate our results, which show that opportunistic species presented the highest M values, in addition to the greatest increases in uncertainty of M (Table 4).

The effects of climate change due to an increase in temperature may vary by region and cause adverse environmental events, such as hydrological, oceanographic alterations, as well as extreme climatic events (Marengo et al., 2009). One reason that would explain the additional natural mortality of freshwater species, which in this study are Amazonian species (Cichla temensis, Prochilodus nigricans and Triportheus angulatus), would be the environmental changes associated with extreme climatic events in the Amazon region, such as extreme flooding and droughts (Freitas et al., 2013). The marine species (Cynoscion acoupa, Lutjanus synagris and Sardinella brasiliensis) may have been affected by impacts on coastal circulation patterns, such as coastal resurgences and South Atlantic central water infiltrations (Garcia et al., 2004; Diaz & Rosenberg, 2008; Gruber, 2011). Compared to marine species, freshwater species are more vulnerable to habitat loss, which can result from geographical barriers and environmental degradation induced by anthropogenic actions, such as hydroelectric dams and pollution (Comte & Olden, 2017). In addition, the thermal tolerance ranges of freshwater species are on average 1 °C greater than for marine species, which probably reflects the greater temperature variability in lakes and rivers and greater stability of the oceans (Dahlke et al., 2020), and which could explain the lower number of uncertainties about M for marine species.

The uncertainties and variability of estimates of M observed in this study are related to the life strategies of each species. Many aspects of the life history of fish are related to climate on an evolutionary scale, which gives them high or low plasticity, i.e., the ability to adapt to natural changes in their habitats (Val & Almeida-Val, 1995). However, the climate changes that are currently underway are occurring at a higher rate than the adaptation rates, and it is necessary to understand, from the aspects of the life history of a species, if the species has the potential to deal with these changes in the environment (Dahlke et al., 2020; Sunday, 2020), since life history traits are the underlying determinants for population responses to environmental forces (King & McFarlane, 2003).

Several studies have been carried out that relate the characteristics of the life history of fish species to temperature in the face of climate change. Studies comparing lake fish from temperate and tropical regions along latitudinal gradients have shown that fish species from lower latitudes are often smaller, grow faster, mature earlier, have a shorter life expectancy, and allocate less energy for reproduction than species from higher latitudes (Blanck & Lamouroux, 2007; Jeppesen et al., 2014; Meerhoff et al., 2012). Jeppesen et al. (2014) conducted a literature review and synthesized the expected changes in the main characteristics of fish life history of inland water systems. In a temperature gradient, species with opportunistic life history characteristics showed greater plasticity compared to species in equilibrium. It has also been shown that the influence of temperature is greater and more significant for fish populations with opportunistic characteristics (faster growth, early age of sexual maturity and shorter life expectancy), than fish with slower life histories, such as species in equilibrium (Free et al., 2019), which was corroborated by our results.

King & McFarlane (2003) analyzed the life history characteristics of 42 commercially important fish species from the west coast of Canada. The authors grouped the species by life strategies and made inferences in responses to environmental conditions for fisheries management purposes. They concluded that equilibrium strategists can withstand only low or moderate rates of capture, since they have low fecundity and late maturation and, therefore, are not able to recover as quickly as other fish after population reduction by fishing and, because they have low growth rates, their population dynamics have a very low variability. As seasonal strategists are relatively long-lived, they could benefit from this by ensuring a relatively long reproductive cycle, which minimizes the risk that periods of unfavorable environmental conditions might result in the loss of a stock. However, these species exhibit lower variability in abundance and were classified as having a steady-state population pattern. Opportunistic species occupy habitats not only with a high degree of variability, but also with large energy resources, so their high population abundance variability.

These studies corroborate our results about increased variability and uncertainty in natural mortality estimates (M) and, consequently, in regard to population dynamics. The increase in temperature has less influence and, therefore, a lower degree of uncertainty of M for strategists in equilibrium (Cichla temensis and Cynoscion acoupa) regarding climate change. The seasonal strategists (Prochilodus nigricans and Lutjanus synagris) presented an intermediate and/or stationary degree of uncertainty of M and the opportunists or with characteristics that come close to the opportunistic strategy (Triportheus angulatus and Sardinella brasiliensis) presented the highest degree of uncertainty of M. This suggests that the adaptation and plasticity of fish species in order to deal with current climate changes, associated with life strategies, follow a gradient of temperature and natural mortality (M), in addition to a general behavior in the variability of the population dynamics (Fig. 5).

Figure 5 Expected behavior according to the characteristics of the life history of fish according to a gradient of temperature (°C) and natural mortality (M) (adapted from Jeppesen et al., 2014).

Each groups of species are described by key life history traits.

The natural mortality (M) is directly related to the productivity of the stock and the yields that can be obtained, in addition to being one of the most sensitive parameters of fish stock assessment models. Therefore, it is essential to use M estimates that incorporate variations and uncertainties in the face of global climate change. In addition, the patterns found in uncertainties about M associated with species groupings by life strategies can be used in more holistic approaches to the assessment and management of recently exploited fishery resources or those with limited biological data.

Supplemental Information

Supplemental Information 1 Supplemental Material - Appendices 1 and 2.

Click here for additional data file.

Additional Information and Declarations

Competing Interests

Author Contributions

Data Availability

The authors declare that they have no competing interests.

Caroline Pereira Campos conceived and designed the experiments, performed the experiments, analyzed the data, prepared figures and/or tables, authored or reviewed drafts of the article, and approved the final draft.

Sandro Dimy Barbosa Bitar conceived and designed the experiments, analyzed the data, prepared figures and/or tables, authored or reviewed drafts of the article, and approved the final draft.

Carlos Freitas conceived and designed the experiments, authored or reviewed drafts of the article, and approved the final draft.

The following information was supplied regarding data availability:

The mathematical concepts and definitions of Fuzzy Set Theory are available in the Supplemental File.

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
