# Peer review of "Uncertainties regarding the natural mortality of fish can increase due global climate change"

_PeerJ, doi:10.7717/peerj.14989_

## Round 0.1 · original submission · Major Revisions

As you can see, two reviewers have commented on your manuscript, finding it interesting and - in general- well-written. Some clarification on the methodological parts is needed, with particular attention to the definition of the fuzzy methods. Also, some additional integration was requested regarding the estimation of natural mortality. I tend to agree with the last suggestion, and I would ask you to carefully consider this request while approaching the revision.

Reviewer 1 ·

Basic reporting

Review comments (other comments and suggestions within enclosed pdf as well),

This is my understanding of what the study tried to do:
The study focuses on the parameter natural mortality M in fish under different IPCC temperature scenarios and compares three freshwater species and three marine species. Instead of using the classical r and K strategists (Pianka 1970), the authors adopt alternative life strategies: equilibrium, opportunistic, and seasonal (following Winemiller 1989) to investigate effects on natural mortality.

Generally good English with some unclarity that I have highlighted in the pdf. I have also suggested some relevant literature that I think could add to the background and importance of the study. The structure of the article is professional. The figures and tables are all relevant, tidy, and aid in understanding the findings. See below for suggestions of things that need clarification and further analysis.

Experimental design

Main concerns:
1. There is some unclarity about the overall aim of this paper since the method seems to provide estimates of future and elevated levels of M while also accounting for how variable environments from IPCC projections may cause variability in M, while the presentation and discussion both focus on uncertainty in M. Since M can potentially fluctuate more due to increasing temperatures, estimation of M can also become more uncertain. I would suggest going through the manuscript and clarify whether it is indeed the estimation uncertainty of M that is of interest, or if it is the variability and level of M that are investigated. It will make it easier to understand if the manuscript is consistent and explicit across all sections. Perhaps a stated definition of what is meant by uncertainty and variability would also be beneficial.
2. As is also mentioned in the manuscript, there are always assumptions made when trying to estimate M, which are uncertain or biased depending on circumstances and population characteristics. Pauly (1980)’s method is a well-known way to estimate M, but there are also other methods, which can be used easily to give more credibility to your findings (especially when there are many uncertainties about M). An easily accessible online tool for estimating M through different methods exists (http://barefootecologist.com.au/shiny_m.html), and I suggest including some of these (if not all) estimation methods to see if the results found in this study are the same with these different methods. It is becoming common practice to use several estimation methods due to the methodological uncertainties around estimating M.
3. It would also be relevant to discuss how increased temperatures might impact asymptotic length (L∞) of fish, which would be an interesting discussion that can connect to a big topic in the literature that has direct bearing on natural mortality (see Gislason et al., 2010). Especially, since size has such a pronounced effect on natural mortality and including how warming may affect asymptotic size and thereby natural mortality (smaller fish, higher likelihood of being eaten, see Scharf et al., 2000) would give the paper broader value. Can also refer to Levangie et al., 2021, since this study explores the mortality consequences of reduced body size hypothesized to result from globally warming water temperatures.
4. The fuzzy method is difficult to understand and generally needs more detail to be replicated. I suggest including a clearly written definition of the fuzzy method in the introduction and mention it again in the method when the concepts are defined more in detail. Also, to make it clear how it relates to other methods for error propagation, including bootstrapping?
o Something that particularly seemed strange to me is the narrow distribution of M before change (figure 5-10) and then broader distributions after all the IPCC scenarios. It seems strange because the only thing driving the model (looking at one species at a time) is temperature, so that seems to show that the temperatures from the IPCC scenarios are more variable than the current temperature (figure 4). It might be helpful for the readers to also see the IPCC distributions also in figure 4 to make the drivers of the model very clear. Another thing to be clarified is why the current range of temperatures are so small when the study area is quite big.
o In relation to this method being some sort of error-propagation, it might also be interesting to look at the sensitivity of variation in other parameters (e.g., temperature, L_inf, and K). However, this might be time consuming, and I suggest focusing on incorporating several methods for estimating M rather than the sensitivity of varying these other parameters due to the focus of the study.

Validity of the findings

I find this a relevant and important study both due to the increasing temperatures and the general uncertainties that persist around the very important parameter M. The study has novelty by separating life strategies into the three life strategies and investigating M estimation for these.

As mentioned above, the method description as of now is tricky to follow and so replication of the study would be difficult for me at least to do.

Data and conclusions are mostly well stated, but see comments for suggestions to what can be improved to make it clearer.

Additional comments

Minor concerns that might be useful to incorporate or adjust to create further relevance or clarity:
- One reference that seems to be relevant both in the introduction and the discussion is Gislason et al. 2010, since they find a significant relation between M and L∞ and found no significance between M and temperature.
- Line 107: The first research question needs to be stated more clearly as it is grammatically not correct and thus ambiguous (reformulation suggestions in pdf).
- Line 115 Fig. 1: I suggest to not include the climatic zones on the map (Fig. 1) as all the data points were collected in the tropical climatic zone, which could just be mentioned in the text and on the figure. That will create more space for the other things, making it easier for the reader to see it.
- Line 141: I would suggest to try and shorten the species description as it seems to be a bit long and detailed.
- Line 210, 255 and 267: Not sure what is being referred to when [80] [2] or [118] are used.
- Line 244: Where does the data come from? Please, make it clear here if they were created or if they were gathered from those points in figure 1.
- Line 253-260: The paragraph “Average Annual Temperature” is unclear. There are eight basins but only three were measured and used in this study, if I understand it correctly. What is meant by “watershed” (highlighted in line 258)?
- Line 256 and 258: In the three basins or watersheds (not sure) is also where growth parameters of all the species in this study were carried out? (Also for marine species or?). Did the authors collect this data (growth parameters and temperature) or did they receive it from somewhere?
- Line 288: The estimation of temperatures is based on IPCC data or collected data or both (mentioning each set of temperature data)? It is unclear what those “sets of temperature data” are coming from.
- Line 297: The numbers in equation (1), where do they come from?
- Line 336: The authors cover many different subjects in the discussion, which makes it difficult to follow their argumentation and explanation of the results. Focus the discussion on fewer things instead of many different things.
- Line 362-368: I wonder whether there is a missed opportunity of comparing the different life strategies across marine and freshwater habitats. Now these are treated more in isolation, but for the reader it would be interesting to hear the authors' take on what effects follow from habitat and which follow life strategy. (Particularly in the discussion).
- Describe the fuzzy set temperature figures 2-4 (or is it maybe one figure with a-c figures?) a bit more. What is membership for example? If they are not one figure, then I suggest having them as one figure with one explanation.
- Line 428: It is unclear how the findings about the uncertainty patterns of M (whether it is estimation or the fluctuation of it) can be used in holistic approaches for assessment and management or for limited biological data species. Seemingly, the study merely found these patterns, which can cause more complications for assessment and management both for data-rich and data-poor species. Thus, investigating how fisheries (potentially increased F) might also increase uncertainties of the stocks’ survival. Perhaps elaborate on the usage of these results and I suggest looking at Plaganyi et al., 2022 for this section as well, as they discuss the value of distinguishing between what causes changes in M (e.g., fishing, predation, or environmental).
- Please double check that you are referring to the right figure in the text (seems to not align in the current format).
- The degree of membership (mentioned in appendix & on figures 2-4) and “set” needs more explanation of what it is in this context.
I hope these comments make sense and are relevant for improving the manuscript further.

Citation for references not mentioned in the submission
Gislason, H., Daan, N., Rice, J.C., Pope, J.G., 2010. Size, growth, temperature and the natural mortality of marine fish. Fish Fish. https://doi.org/10.1111/j.1467-2979.2009.00350.x
Levangie, P. EL, Blanchfield, P.J., Hutchings, J.A., 2021. The influence of ocean warming on the natural mortality of marine fishes. Environ. Biol. Fishes. https://doi.org/10.1007/s10641-021-01161-0
Plaganyi, E.E., Blamey, L.K., Rogers, J., Tulloch, V.J.D., 2022. Playing the detective : Using multispecies approaches to estimate natural mortality rates. Fish. Res. 249, 1–18. https://doi.org/10.1016/j.fishres.2022.106229
Scharf, F.S., Juanes, F., Rountree, R.A., 2000. Predator size - prey size relationships of marine fish predators : interspecific variation and effects of ontogeny and body size on trophic-niche breadth. https://doi.org/10.3354/meps208229

Annotated reviews are not available for download in order to protect the identity of reviewers who chose to remain anonymous.

Reviewer 2 ·

Basic reporting

The article is written in clear English, has sufficient references, and relevant hypotheses. The authors project is based on simulations and therefore describe their model and parameters used within the text. However, they did not provide code or describe the computational methods (version of Python, key packages, model specifications such as number of iterations) that would help clarify important details about their methods.

Experimental design

The authors conducted an interesting and important study about how a key population growth parameter may be affected by a warming climate, and how this parameter varies across life history strategies and regionally important species. This information will be important for a more holistic understanding about population dynamics amidst a more volatile future. The question is well-defined and investigated. The authors provide information to re-create the model in the Materials and Methods but do not provide code or details about coding (e.g., main packages used) to further replicate the study. Moreover, there is limited information about model specifications that would greatly aid reproducibility.

Validity of the findings

The findings are important, impactful, and provide baseline estimates for a key population growth parameter under an uncertain future. The methodology and underlying data appear to be sound and robust, however additional model details and description are needed in the Materials and Methods to increase the project's reproducibility. The conclusions of the article are well-stated, though some re-structuring of the Discussion section to emphasis key points earlier would likely enhance readability. Overall, beyond these two concerns about the project that I think need to be addressed, I commend the authors for their hard work and interesting study.

Additional comments

Line 57. Perhaps phrase as “related to the survival of many ecothermic animals, especially fish (Baudron et al., 2014).”

Line 71. Perhaps state “smaller species in lower trophic levels” to consider that trophic levels are not necessarily static and a fixed attribute of species.

Line 103-106. Perhaps rephrase as “In this study, we assumed two points: (i) whether the estimate... ; and (ii) whether the amplitude…. species-specific (Ropke et al., 2017). We then answered two questions: (i) Have… ” This suggestion is because, in my opinion, the sentence is long and this helps break it into two sentences.

Introduction: Overall, this is a wonderful introduction that provides ample background material in a cohesive manner to address key questions about the estimation of key population parameters under global change.

Line 130: Perhaps change “prevail” to “persist” to more directly relate it to population persistence and potential local extirpation.

Lines 160-161; 176-178; 196-198; etc. Regarding the species parameter lists listed at the end of their respective paragraphs, I think it is important for you to state whether you estimated these and made them relative to the other species of interest, or obtained most of these data from the previous list of natural history information (e.g., growth rates but not theoretical maximum length). I think it would be helpful to state once, potentially at the end of the first species description, how you did this. For example, “We used these natural history data to approximate model parameters for major life history characteristics for each of the target species. We considered C. temensis to have X, Y, and Z.” After stating it for the first species, however, I don’t think it would be necessary to reaffirm it in each paragraph. All that said, I really enjoyed and appreciated the natural history details, particularly in relation to local and regional commercial use.

Lines 244-249. I would recommend separating this into two sentences: one for the input variables, and a second for the forcing variables.

Line 250-251: This single sentence paragraph might be better consolidated with similar information. You could potentially move this information into the species descriptions (comment about Lines 160-161) and refer to Table 1 earlier so this information is accessible as readers assess the species.

Line 290: Potentially change to “For each RCP scenario”, as you described the connection between IPCC and RCP earlier, and fewer acronyms can help readability.

Line 299: I would recommend stating the version you used, as well as any major packages that you used. This is helpful so that others can replicate your study from a computational standpoint.

Line 301-334: I would reference Figures 2, 3, and 4 when you discuss the temperatures across basins. Currently, you just have a reference for Figure 2.

Lines 307: I would state in the first sentence that these are the freshwater species. Similarly, I would state in the first sentence of the following paragraph that these are the saltwater species. This would help the reader understand the context of each paragraph more fully.

Lines 308: Are scenarios C1 and C4 the 4 different RCP scenarios? After a search that seems so. I think it might be more intuitive for the reader if you simply stated the RCP scenario rather than this new abbreviation.

Lines 320-322: I would simply add a reference to Table 4 in the freshwater and saltwater fishes paragraphs so that readers can go to the results after reading them instead of a separate paragraph for that.

Line 325: I don’t think you need to reiterate that IPCC scenarios imply temperature increases more than once in the results. The way you state it in the previous sentence is great.

Lines 327-334: These are very interesting results and key findings for many stakeholders in fisheries. I think it is very well written and explains the main points concisely. Perhaps you could elaborate a little about the last point, that freshwater fish have a greater degree of uncertainty. For example, because these M estimates are based on temperature projections, I find it interesting that the Amazonian River basin has the smallest range of temperature change (Lines 302-303), yet greater degree of uncertainty. This suggests smaller changes in temperature (shallower range) have much larger consequences on M for freshwater fish.


Lines 351-354: I would recommend re-wording this section. I think all of these links between ecological and environmental factors are largely true, but it is written such that it kind of suggests these are results from your particular study. However, you did not model dissolved oxygen, and you did not model population dynamics. Therefore, I recommend rewording it to state that, in many cases, these are the consequences of increasing temperatures on fish populations, though exceptions exist.

Line 354: Add “)” after 2013 in “Barros & Albernax, 2013.”

Lines 355: This is an interesting finding and comparison to make. Maybe separate this into 2 sentences. Something like, “The allometric scale of metabolism (Woodward et al., 2016) would suggest that larger species with longer lifespans and slower life cycles (i.e., equilibrium species), such as … would have higher local extinction risks. Yet, our results suggested that opportunistic species exhibit the greatest increase in uncertainty of natural mortality. (Table 4.)” In addition, after reading the remaining discussion, I think this section is out of place. It might be better to discuss how your results corroborate numerous studies (the following paragraphs) prior to highlighting one specific contrasting pattern that, in light of your discussion about life history strategies, physiology, etc., clarifies why this discrepancy exists.

Lines 370-375: I don’t think you need to state the temperature increases for each scenario and re-explain what the concentration scale implies.

Lines 379-386: I would recommend removing the details about temperature increases associated with RCPs but retain and emphasize the Dahlke et al., 2020 section. This paper, which stated most fish are at risk under large temperature increases, could be relevant to several other parts of your discussion. For example, you could talk about the risk for freshwater and marine, based on Dahlke, at the end of the preceding paragraph.

Lines 394-395: I am having a hard time following “the influence of temperature is greater and more significant, both positively and negatively, for fish populations”. Could you clarify what you mean by this?

Lines 410-413: I think you can remove this and simply add something about high population abundance variability at the end of the previous sentence.

Lines 414: Consider rephrasing to “These studies corroborate our results about increased variability and uncertainty in natural mortality estimates (M) across…”

Line 423: Considering adding “(Fig. 5)” in place of “, as can be seen in Fig 5.”

Table 4: Consider changing C1-C4 to RCPs for clarity. If you want to keep using Cs, I would recommend clearly stating that at the end of the introduction or in the methods so it isn’t a surprise in the results.

Figures and Tables: I would double-check that these are all referenced in order and appropriately throughout the text.

---

## Round 0.2 · Minor Revisions

The reviewers suggested that you did a great job with the revision, and the manuscript is almost ready for publication. As you can see, however, Reviewer 1 still reported some very specific comments and issues (mostly typos or requests for clarifying some sentences) that I think can be addressed very quickly.

Reviewer 1 ·

Basic reporting

I reviewed also the original submission of this study. The text has thoroughly been improved and it is easy to follow the study as is. The study is comprehensive and interesting.

Experimental design

Figures and tables work as good support along with the text making it easy to maintain an overview over what was done. Improvements have been done on explaining the fuzzy methods.

Validity of the findings

Particularly, the discussion has been improved making it easy to follow while also interesting points and relevant studies are mentioned.

Additional comments

Line 62: Not clear to me what is meant by this sentence.
Line 69: is the reduction of predators referred back to the sentence before when describing large size and high longevity?

Line 101: What is meant by assuming two points? These sound more like questions rather than assumptions made.
Line 105: “i) Will the temperature increases […] increase uncertainties …”

Line 118: Not also marine species, or?

Line 247: I think you mean “we use”.
Line 249: Not sure I understand what is meant by “force variables in the modeling”.

Line 286: Could the fuzzy average annual temperatures estimated be included in one of the tables? Maybe in table 3.

Line 293: What is m and M in the interval?

Line 289 & 297-299: These two sentences confuse me. Are you estimating values of fuzzy average annual temperature per IPCC scenario but also introducing the variations from IPCC on to these values?

Line 303: misspelling “will”.

Line 323: In figure 2 it looks like the temperature range is only relevant where the spike is, meaning that the temperature range looks to be between ~27-27.8 degrees as the “membership” is 0 for the rest. Maybe I am interpreting it wrong.

Line 328: “… a pattern of increased uncertainty of M with increasing temperature…”.

Line 360-362: Unsure about what is meant by “baseline natural mortality” here.

Line 369: “…increasing to around 4…”

Line 376-377: The sentence is written in Portuguese, I think

Reviewer 2 ·

Basic reporting

This article is a valuable contribution to the literature by demonstrating the uncertainty and suspected increase in natural mortality estimates of fish. This information is of great importance given the likelihood of sustained temperature increases and key role of fish in maintaining ecosystem services and functionality. This paper passes all three noted sections of the PeerJ standards and, in my opinion, should be recommended for acceptance after the submitted revisions. I am particularly impressed with the amount of details added to the paper, particularly the methodology, and commend the writers for filling this gap.

Experimental design

No comment

Validity of the findings

No comment

---

## Round 0.3 · accepted · Accept

Thank you for your fast reply: I think the manuscript is now ready for publication. Congratulation!